# Multifunctional Role of S100 Protein Family in the Immune System: An Update

**DOI:** 10.3390/cells11152274

**Published:** 2022-07-23

**Authors:** Parul Singh, Syed Azmal Ali

**Affiliations:** 1Cell Biology and Proteomics Lab, Animal Biotechnology Center, ICAR-NDRI, Karnal 132001, India; parulbiotech92@gmail.com; 2Division of Proteomics of Stem Cells and Cancer, German Cancer Research Center (DKFZ), 69120 Heidelberg, Germany

**Keywords:** nutritional immunity, inflammation, immune cells, alarmins, antimicrobial peptide, autoimmune disease, COVID-19

## Abstract

S100 is a broad subfamily of low-molecular weight calcium-binding proteins (9–14 kDa) with structural similarity and functional discrepancy. It is required for inflammation and cellular homeostasis, and can work extracellularly, intracellularly, or both. S100 members participate in a variety of activities in a healthy cell, including calcium storage and transport (calcium homeostasis). S100 isoforms that have previously been shown to play important roles in the immune system as alarmins (DAMPs), antimicrobial peptides, pro-inflammation stimulators, chemo-attractants, and metal scavengers during an innate immune response. Currently, during the pandemic, it was found that several members of the S100 family are implicated in the pathophysiology of COVID-19. Further, S100 family protein members were proposed to be used as a prognostic marker for COVID-19 infection identification using a nasal swab. In the present review, we compiled the vast majority of recent studies that focused on the multifunctionality of S100 proteins in the complex immune system and its associated activities. Furthermore, we shed light on the numerous molecular approaches and signaling cascades regulated by S100 proteins during immune response. In addition, we discussed the involvement of S100 protein members in abnormal defense systems during the pathogenesis of COVID-19.

## 1. Introduction

S100 is a large subfamily of low-molecular weight calcium-binding proteins, consisting of numerous isoforms (30 isoforms in humans) with structural similarity and functional differences. In 1965, Moore designated this protein as “S100” due to its solubility in 100% ammonium sulfate at neutral pH [1]. The S100 protein family is unique among all other Ca^2+^-binding proteins in terms of its structure, molecular conformation, functions, and on account of accessibility as extracellular and/or intracellular proteins [2]. Due to their existence in only vertebrates, S100 protein family were determined to be phylogenetically new proteins. The whole-genome sequence analysis of invertebrates, including plants, drosophila, yeast, and nematodes such as *C**. elegans*, revealed no S100 family protein expression [3,4]. In humans, there are 24 S100 isoforms in the epidermal differentiation complex (EDC) cluster on the chromosome locus 1q21. Additional S100 isoforms have been identified at various chromosomal locations, including S100B (21q22), S100G (Xp22), S100P (4p16), and S100Z (5q14) [5].

The S100 protein monomer consists of two helix–loop–helix structural motifs, and is also known as EF-hands. These two EF-hands contain the binding potential for transition metal [6], are arranged consecutively, and are connected through a flexible hinge region in the center [7,8,9,10]. For complete S100 protein activation, it requires two mechanistic regulatory steps. The first is transition metal binding (Ca^2+^, Zn^2+^, Cu^2+^, and Mn^2+^) [11,12] for its folding. The second is the formation of homo- and heterodimers [13]. S100 isoforms show high structural similarity. However, variation in the C-terminal extension and hinge region causes sequence variability, which could be a reason for functional discrepancy among members [14].

S100 proteins can act intracellularly and extracellularly. Intracellular S100 isoforms control immune system functions, such as transcription regulation, trafficking activity, intracellular receptors, free radicals scavenger, and cytoskeleton rearrangement, to name a few examples. Moreover, secretory S100 isoforms can interact with a variety of cell surface receptors, including G protein-coupled receptors (GPCR), Receptor For Advanced Glycation End-Products (RAGE), Toll-Like Receptor-4 (TLR-4), proteoglycans heparin sulphate and N-glycan, and scavenger receptors [15] (Figure 1). S100 proteins, in particular, function as cytokines and bind with RAGE and TLR-4 to activate the pro-inflammatory signaling cascade, thus increasing immune cell recruitment for their proliferation and differentiation. This protein also enhances the expression of MMPs (Matrix *metalloproteases)* and CAMs (*Cell* adhesion molecules) required for tissue remodelling and chemotaxis, respectively. Notably, S100 protein members undergo extensive post-translational modifications to acquire functional activity; the isoforms and modifications are as follows: S100B, S100A1, S100A8 (nitrosylation), S100A8/A9, S100A11 (phosphorylation), S100A8/A9 (carboxymethylation), S100A3 (citrullination), S100A11 (transamidation), S100A14 (myristoylation), S100A1 (glutathionylation), S100A8/A9 (oxidation), and S100A4 (sumoylation) [16].

S100 protein family members play a wide range of roles in healthy cells that are not restricted to calcium storage and transport (calcium homeostasis). Instead, they extend to diverse cellular activities, such as S100A8/A9 scavenging ROS and nitric oxide (NO). S100A1, S100A4, S100A6, and S100A9 monomers are involved in cytoskeleton assembly. S100A10 and S100A12 function in membrane protein recruitment and trafficking. S100A4, S100A11, S100A14, and S100B are responsible for controlling transcriptional regulation and DNA repair. S100A6, S100A8-A9, and S100B are involved during cell differentiation. S100A8-A9, S100A12, and S100A13 are involved in the release of cytokines and antimicrobial agents. S100A1 is involved in muscle cell contractility, while S100A4, S100A8/A9, S100B, and S100P are involved in cell growth and migration. Finally, S100A6, S100A9, and S100B all play a role in programmed cell death [17]. 

Separate genes for each isoform in different chromosomal locations, as well as their existence as intracellular and extracellular proteins, enable S100 proteins to perform a wide range of functions. Additionally, the ability to bind with different transitional metals (Ca^2+^, Zn^2+^, and Cu^2+^), the ability to form non-covalent homo- and heterodimers, and significantly less sequence homology in the hinge and extended C terminal region, all contribute to the functional discrepancy of this protein group [11,18,19].

This article highlights the multi-functional role of S100 protein members associated with the immune system. The immune system is mostly controlled by three processes and/or components that fall into two categories: innate immunity and adaptive immunity. These components include immune system cells (myeloid and lymphocytes), active molecules (alarmins, antibodies, cytokines, interleukins, chemo-attractants, antimicrobial peptides, and components of the complement system), and the immune process (inflammation, complement system, phagocytosis, and necrosis). S100 protein family members have the potential to function as active immune system molecules. In this review, we will discuss how members of the S100 protein family participate in a variety of active immunological and associated responses.

## 2. Function of S100 Protein in Host Defense Mechanism

The host defense mechanism, or immune system, is a uniquely managed systematic process in mammalian physiology. It is not limited to higher organisms, but is instead employed by all life forms to counter invading unwanted microbes. Certain microorganisms, such as viruses, act as pathogens, inducing the host’s defense system. However, all microbes are not pathogenic; some are beneficial, or good, bacteria (e.g., intestinal microbiota or probiotics) [20]. For instance, healthy symbiotic microbiota are involved in digestion, anti-inflammatory, anti-infectious, and immune-modulating processes [21].

On the other hand, harmful invading microbes impair normal bodily functions, resulting in severe illness, and, if left untreated, death. The immune system uses both a fundamental (adaptive immune response) and a non-specific defense strategy in order to fight infection (innate immunity or first line of defense) [22,23]. The immune system utilizes various strategies to defeat pathogens, including (I) master processes such as inflammation, complement system, phagocytosis, necrosis, and apoptosis (II) immune cells such as monocytes, macrophages, neutrophils, natural killer cells, dendritic cells, and lymphocytes (III) communication through molecular factors such as antibodies, cytokines, interleukins, chemo-attractants, antimicrobial peptides, PRRs, and other proteins. Interestingly, S100 family proteins are important in all three above-mentioned immunological functions, such as in the regulation of immune cells, and operate as active molecular factors in significant immunological responses or diseases. 

### 2.1. Role of S100 Protein in Most Prevalent Innate Immune Cells

#### 2.1.1. Macrophage and Monocytes 

Macrophages are key players in the role of an active immune system. Bacterial LPS (lipopolysaccharides), also known as lipoglycans and endotoxins, cause macrophage activation. This promotes S100A8 monomer expression by activating the TLR-4 receptor on macrophage surfaces [10]. TLR-4 activation amplifies the signal via the downstream signaling cascade, activating various transcriptional factors, such as NF-κB, AP-1, and IRF-3, via non-endosomal and endosomal TLR-4 pathways [24]. These transcriptional regulatory factors regulate the transcription of primary response genes (IL-10), as well as class II transcriptional factors (C/EBPs, AP-1, and Stat-3). Further, in macrophages, IL-10 increases the expression of S100A8 monomer. Stress-induced ROS through NADPH oxidase (NOX) and mitochondria also induces S100A8 monomer expression during pathogen perturbation, implying that S100A8 functions as a stress response element. The intracellular S100A8/A9 heterodimer acts as an oxidant scavenger by binding to cytoskeletal proteins, to rearrange the cytoskeleton, and releasing into the extracellular matrix through non-classical secretory pathways [10] (Figure 2). In addition, the induction of S100A8, S100A9, and S100A12 heterodimer complexes in macrophages is a complicated process that is heavily influenced by proinflammatory factors. LPS, for example, induces the expression of the S100A8 monomer in endothelial cells, fibroblasts, and macrophages, which can be amplified by glucocorticoids [25]. However, IL-10 has no direct influence on the expression of the S100A8 /S100A9 heterodimer complex. Instead, Th2 cytokines, such as IL-4 and IL-13, can suppress S100A8 /S100A9 heterodimer production in macrophages generated by LPS [26,27].

CD147 is an EMMPRIN (extracellular matrix metalloproteinase), or basigin, a transmembrane protein that is abundantly glycosylated and serves as an inducer of extracellular MMPs in various cell types, including hematopoietic and leukocyte cells. Current research shows that CD147 can bind to the spike protein of COVID-19, and may be involved in the invasion of host cells [28,29]. Another protein, CyPA, is a known EMMPRIN ligand, and is required for monocytes/macrophages to regulate MMP-9 and chemotaxis [30]. S100A9 stimulates the release of pro-inflammatory cytokines by binding to the TLR-4 receptor and activating the NF-κB transcription factor, resulting in the expression of pro-inflammatory response genes in monocytes (Figure 3). A recent discovery indicates that S100A9 is involved in monocyte/macrophage migration during the pro-inflammatory process. Similar to CyPA, monocyte and macrophage chemotaxis, via S100A9, is selectively dependent on EMMPRIN. However, migration via the S100A8/A9 heterodimer is independent of EMMPRIN. S100A9 primarily induces ERK and Akt phosphorylation by interaction with EMMPRIN, promoting monocyte and macrophage migration via an EMMPRIN/ERK-dependent pathway [31]. It can be concluded that EMMPRIN only participates in the momentary action of monocytes/macrophages via the S100A9/A9 homodimer, but does not participate in S100A9 monomer- or S100A8/A9 heterodimer-induced inflammation and chemotaxis of macrophages/monocytes. S100A8 and S100A9 also improve monocytes’ ability to perform their functions as Ca^2+^ stores/sensors, as well as Ca^2+^-dependent interactions with the cytoskeleton, enhanced movement, increased degranulation, increased phagocytosis, S100A9 monomer downregulation, and microtubule polymerization [32].

S100A12 expression is higher in classical (CD14^hi^CD16^-^) monocytes than in non-classical (CD14^+^ CD16^hi^) monocytes, and decreases during monocyte-to-macrophage differentiation, but not during macrophage polarization, according to some studies. Additionally, S100A12 expression is modulated by monocytes in periodontitis. This altered level of S100A12, in both peripheral circulatory and gingival tissue monocytes, indicates its functional role in periodontitis pathogenesis. Therefore, it can be concluded that S100A12 is primarily expressed and released by monocytes, rather than by differentiating macrophages. Furthermore, the accumulation of S100A12 in inflamed tissue indicates that it is initially released from monocyte cells [33].

#### 2.1.2. Neutrophil 

Several members of the S100 family, including S100A4, S100A6, S100A8, S100A9, S100A11, and S100A12, have been found to be expressed in neutrophil cells [34]. The expression profile of each isoform is distinct; for instance, S100A8 and S100A9 are expressed abundantly, whereas S100A4 is constitutively expressed, and S100A6 and S100A12 expressions are restricted or conditional [10]. Differential expression of isoforms is depending on distinct stimuli; for example, physical damage, such as injury or UV irradiation, induces S100A8 and S100A9 expression in keratinocytes [28]. The expression of these isoforms in different immune cells can be affected by PAMPs (pathogen-associated molecular patterns) such as LPS, double-stranded RNA, and bacterial flagellin protein. Similarly, the pro-inflammatory cytokines TNF-α and IL-1β promote calgranulin (S100A8, S100A9, and S100A12) upregulation in keratinocytes and microvascular endothelial cells. It is important to note that, due to the antimicrobial activity of S100A8 and S100A9, these S100 proteins are also referred to as calprotectin [27].

Extracellular S100A8/A9 heterodimer release is essential for enhancing inflammatory responses via aberrant regulatory activity, either autocrine activation of neutrophils or paracrine stimulation of other inflammatory cells [28,35]. In addition, S100A8 and S100A9 proteins promote phagocytosis and increase ROS levels. Despite this, S100A8 inhibits ROS and Ca^2+^-dependent cytoskeleton–cytoskeleton interactions, leading to increased migration, degranulation, and phagocytosis. As a result, S100A9 inhibits microtubule polymerization, whereas S100A12 regulates neutrophil Zn^2+^ homeostasis [32]. Hence, S100A8/phospho-A9, but not the S100A8/A9 heterodimer, regulates the expression of cytokines (IL-1α, IL-1β, TNF-α, IL-6) and chemotactic factor, including CCL2 (monocyte attraction), CXCL8 (neutrophil attraction), and CCL3 and CCL4 (NK cell attraction) [35]. Furthermore, the mechanism of S100A8 and S100A9 secretion from various cells is dependent on the type of stimuli. Normally, S100A8 and S100A9 are secreted when an activated monocyte interacts with endothelial cells. However, dead cells can also stimulate neutrophils to secrete S100A8 and S100A9 [35] (Figure 4).

Activated neutrophils induce chromatin decondensation, nuclear membrane disruption, and chromatin release during NETosis [36,37] (Figure 4). NETs (neutrophil extracellular traps) capture microbial invaders, such as fungi and bacterial pathogens, and may have evolved to trap critical microbes that are difficult to consume by phagocytosis. The process is primarily characterized by the release of chromatin, which consists of extended chromatin fibers that intersect and bundle with one another, forming a mesh-work or trap that immobilizes extracellular microorganisms, and is thus referred to as a NET [35]. In vitro induction of NETs has recently demonstrated functional involvement of the S100 protein group. Neutrophil activation is triggered by stimuli such as *Aspergillus fumigates* or *Aspergillus nidulans*, PMA, or MSU, and releases NETs through a mechanism involving NADPH oxidase, myeloperoxidase (MPO), NE, and PAD4. During this step, phosphorylated S100A8/A9 heterodimers are secreted into the extracellular space by neutrophils. Consequently, S100A8/phospho-S100A9 establishes the surrounding neutrophils by energizing them for TNF-α and IL-6 (cytokines) secretion and CCL2, CCL3, CCL4, and CXCL8 (chemokines). The secretion of these chemokines and cytokines is primarily regulated by the TLR-4 and RAGE receptors (participate in CCL2 secretion) over the surface, locally or nearby accessible neutrophils at the inflammatory site. This mechanism indicates that the S100A8/A9 heterodimer plays a significant role in the progression of the inflammatory process [35].

### 2.2. S100 Protein’s Role in the Immune System as an Active Molecule

#### 2.2.1. Alarmins or DAMPs

Alarmins are endogenous molecules that belong to the DAMP (damage-associated molecular patterns) family, which also includes extracellular hyaluronan fragments produced by tissue injury, as well as intracellular heat shock protein and HMGB1 (high-mobility group box 1) [38,39,40,41]. In the aftermath of trauma and microbial infections, alarmins serve as intermediate signaling mediators for the inflammatory process. During a threat, alarmins send an intracellular defense signal to the host defense system’s immune cells by interacting with chemotactic factors and PRRs (Pattern Recognition Receptors, such as TLR, NLRs (NOD-like receptors), and MRs (mannose receptors)) [42,43,44,45]. PRRs enable the innate immune system to detect tissue injury by sensing mislocalization and changes in endogenous effector molecules, such as DAMPs [46]. S100 protein is released via the same route as DAMP [47]. However, the binding of a few S100 protein members to TLR, and the similar secretion pattern, make this protein vulnerable to functioning as an alarmin. 

During tendinopathy, the S100A8/A9 heterodimer (calprotectin) has an immunomodulatory effect that stimulates the innate immune response and controls the stromal microenvironment. In tendinopathy, calprotectin serve as alarmins, recruiting immune cells to nearby areas. As a result, the HMGB1 alarmin stimulates inflammatory cytokine expression and modulates the matrix through TLR-4 dependent receptors in tendon cells. Furthermore, constitutive expression of the S100A8/A9 heterodimer dominates the downstream signaling cascade to enhance the expression and secretion of CCL2, CCL20, CXCL10, IL-6, IL-12, and IL-8 in the tendon matrix, through the DAMP receptor [48]. Moreover, secretory S100A8 and S100A9 amplify the recruitment of immune cells to the tendon matrix; altogether, damage triggers the release of CCL2 from the tenocyte, which recruits monocytes to the inflamed area. During tendinopathy conditions, the inductive expression and secretion of S100A8 and S100A9 affect the activation of local tenocytes, boosting immune cell recruitment to the injured site and altering the stromal microenvironmental cue [48]. Quantifying the expression study of alarmin in diseased supraspinatus tendons suggested that S100A9 and HIF-1α may have pro-inflammatory effects in tendon disease, nuclear IL-33 may protect against pro-inflammatory stimuli, and HMGB1 may play a role in tendon recovery [49]. Alarmins are not limited to tendinopathy; they are much more. T. *gondii* infection triggers the biochemical release of endogenous effector molecules, such as IL-12 and CCL-2, from immune cells in mice and humans. Monocytes can identify S100A11 as an alarmin secreted by parasite-infected cells and activate an innate immune response [50].

Sepsis is a dangerous sickness characterized by hyperactivation of the host defense inflammatory mechanism, which causes activation of the inflammatory process in the entire body and could lead to septic shock or death. Recently, Ulas et al. revealed the role of the S100A8/A9 heterodimer, describing significant function in overcoming sepsis conditions in neonates. They elucidated that S100A8 and S100A9 alarmins control and modulate the reprogramming of genes, such as MyD88-dependent proinflammatory genes, in the perinatal stage of the newborn, to prevent hyper-inflammation without affecting host-pathogen defense mechanisms or TIR-domain-containing adapter-inducing interferon-β (TRIF)-dependent regulatory genes (at the very beginning these are epigenetically silent, but expression increases gradually in the first year of the neonate). Although, alteration in transient S100-mediated gene programming could lead to hyper inflammation and sepsis. Primarily, the S100 protein precisely activates p65, NF-κB, and IRF5 to enhance the expression of pro-inflammatory cytokine genes, leading to the recklessness of the MyD88 signaling cascade. After birth, the direct exogenous insertion of S100A8/A9 heterodimer alarmins shuts out tissue damage, secondary microbial overgrowth, and hyper-inflammation after an *S. aureus* challenge. S100A8 was found to be more effective than the S100A8/A9 heterodimer in shutting off hyper-inflammation and microbial growth [51]. Thus, the S100A8/A9 heterodimer plays a critical role as an immune regulator to hamper extensive inflammation in infants. In brief, a human mother’s breast milk contains a high amount of calprotectin, which mediates sepsis protection during the initial period of a newborn’s life. In vitro studies showed that breast milk calprotectin could inhibit the growth of numerous bacteria, such as *Staphylococcus aureus*, *B streptococci*, and *E.coli*, by withdrawing the essential nutrient magnesium and starving them for the nutrition of newborns [51,52,53].

Apart from microbial pathogen assault, an uncommon, but potential, cause of inflammation or sterile inflammation could be a misregulation of the adaptive mechanism of the defense system. Microbial agents do not always activate an inflammatory process, in simpler terms, instead triggering endogenous factors released as a stress signal or sterile inflammation. The best example of sterile inflammation is calprotectin, a highly abundant alarmin that is secreted during psoriasis, allergies, arthritis, infections, autoimmune disease, pulmonary and heart disease, and intestinal disease. It necessitates an understanding of the systematic molecular approach to causing sterile inflammation. A recently classified mechanism of auto-inhibitory activity of extraordinary calprotectin has been revealed. In auto-inhibition, the Ca^2+^-dependent tetramerization ability of calprotectin exterminates the proinflammatory potential of the S100A8/A9 heterodimer. In short, S100 alarmin-driven inflammation is potentiated to establish self-control. Intracellular Ca^2+^ deficiency typically promotes S100A8/S100A9 heterodimerization and the release of activated neutrophils [54].

#### 2.2.2. Functional Implication of S100 Protein as Antimicrobial Peptides and in Nutritional Immunity

The family of S100 proteins is a critical connecting link in innate immunity. It facilitates the immune response cascade through direct participation, and provides host defense mechanisms by triggering immunological responses against numerous invasion pathogens. Notably, antimicrobial peptides and/or proteins (AMPs) play an essential role in the first line of defense against a wide range of pathogens [55]. In humans, there are many antimicrobial peptides (AMPs), bactericidal factors, and host defense peptides (HDPs), including RNase7 [56], Reg3 [57], α- and β-defensins [58], S100A7 (psoriasin) [59,60], S100A15 [61], the S100A8/A9 heterodimer (calprotectin) [6], and Cathelicidin/hCAP-18 (cleaved into LL-37 and FALL-39) [62,63,64,65]. Keratinocytes also express the S100 protein subgroup (S100A7, S100A8, S1009, and S100A15), which functions as an anti-viral peptide. Thus, keratinocytes show antiviral and immunomodulatory properties through the S100 subgroup, which affects viruses’ replication cycle or activity [66].

The antimicrobial S100 protein (S100A7, also known as psoriasin) was found to be highly expressed in inflamed psoriatic skin or, more commonly, in healthy skin. S100A7 also acts as a chemotactic factor for immune cells, stimulating cell proliferation or differentiation, triggering cytokine/chemokine synthesis (immunomodulatory), and enhancing first-line defense by maintaining skin protection barriers. Both S100A7 and S100A15 exhibit bactericidal activity against *E. coli*. On the other hand, the S100A8/A9 heterodimer and S100A12 act efficiently to defend numerous viruses or fungi categories from invaders [67,68]. Another example of S100 protein’s antimicrobial ability is the high abundance of FLG-2 (S100 fused type protein) protein in the upper epidermis against *Pseudomonas aeruginosa* and other Gram-negative bacteria from soil and water. C-terminal FLG2 fragments act as antimicrobial defense shields by hampering bacterial replication and restricting their growth in the epidermis. The antimicrobial activity of FLG2-4 does not resemble pore formation by insertion. Instead, FLG shows engagement with the cytosolic side of the membrane and impedes replication machinery by hampering DNA polymerase activity, causing bacterial death [69].

S100 as AMPs did not originate in the human defense system; thus, they evolved before mankind, and the best example is bovine. Proof of this is bovine S100A12, which has been potentiated to inhibit microbial growth (*E.*
*coli*) in vitro, suggesting the S100 protein’s capacity to work as an antimicrobial protein. Transcriptional upregulation of bioactive innate immune proteins (S100A7, S100A9, S100A11, and S100A12) [70,71] has been detected in the milk of mastitis-infected mammary glands, compared to healthy ones [72]. Furthermore, S100 protein can bind with the outer surface of the bacterial membrane, through negative-charge phospholipids, to facilitate destabilization and pore formation in the microbial membrane to destroy bacteria, resembling the functionality of the complement system in an efficient arm of the innate immune system [73,74].

The functional implications of AMPs are limitless, not just restricted to pore formation, and include scavengers. For instance, almost all pathogens usually require a surplus of transition metals as nutrients for their growth. In response to infectious invaders, the host’s innate immune system dwindles the essential ions available to starve the microbes, consequently decreasing the pathogen’s growth. This process is called **nutritional immunity**. Calgranulins have the highest expression in infectious conditions, and play a critical role in the innate immune response to restrict microbial growth [67]. S100 protein members can also bind with a transition metal; calgranulins, in particular, take advantage of this intriguing property and inhibit microorganism growth by essential-nutrient deprivation [75]. Another example of nutritional immunity is that of birds and reptiles. Calgranulin (also known as MRP126) promotes the existence of an innate immune response against microbial pathogens in birds and reptiles. Avian MRP126, similar to human calgranulin, can selectively sequester Zn (II) and limit its availability, thereby limiting pathogen-invasion growth [76].

Furthermore, granulocytes (neutrophils) and phagocytic cells first reach the site of infection, govern microbial infection by phagocytosis, and simultaneously initiate various innate immune responses by producing antimicrobial peptides or protein NETosis formation and ROS and NO intermediates. Interestingly, calprotectin is an essential candidate for nutritional immunity, constituting 60% of neutrophil cytoplasm protein content. Neutrophil participates in nutritional immunity by producing calprotectin and innate immune responses via antimicrobial peptide formation (such as calprotectin and lactoferrin) [73]. For example, a broad range of research suggests that calprotectin functions as an antimicrobial protein via metal-chelating capacity, which causes essential ions to be in poor condition for a variety of pathogens such as *Candida albicans, Acinetobacter baumannii, Klebsiella pneumoniae, H. pylori, E. coli*, and *S. aureus*. Calprotectin also regulates the pursuit of proinflammatory virulence factors secreted by them [77]. Moreover, calprotectin obstructs iron uptake and facilitates iron starvation through sequestering Fe (II) at the His6 amino acid position in response to *Pseudomonas aeruginosa* [78]. Similarly, calprotectin also acts as a manganese sequester against *Staphylococcus aureus* [79]. 

S100A7 also acts as an antimicrobial protein, shows bactericidal activity, and inhibits the growth of *E. coli* by Zn-ion depletion through sequestering Zn (II) [80]. However, the R. temporaria protein RtS100A7, a human S100A7 orthologue, lacks a Zn binding site, potentially limiting microbial growth under Zn starvation independently, implying that antimicrobial function evolved early in tetrapod evolution [80].

The rarest example is corneal abrasion (CA), which is an eye injury due to a scratch on the cornea’s surface. Topical insertion of cationic antimicrobial protein enhances resurfacing by replacing damaged cells with new epithelium, or re-epithelialization, at the injury site in corneal abrasion, and facilitates wound healing. During CA, increased transcriptional expression of S100A9 occurs in the cornea, followed by a release into extracellular space, which enables the inflammatory response to defend against invader microorganisms. The S100A8/A9 heterodimer discloses its pro-inflammation cascade function via RAGE and TLR-4 [81]. 

*Helicobacter pylori* are spiral-shaped, Gram-negative bacterium that tenaciously colonize the stomach in about half of the world’s population. Its existence in the gut can cause adverse health consequences, such as peptic and duodenal ulcers, gastritis, MALT (Mucosa Associated Lymphoid Tissue) lymphoma, and invasive gastric cancer. The primary wrongdoer responsible for pathogenesis is cag Pathogenicity Island (cag PAI), which contains genes coding for a secretory effector protein (CagA) and multiple T4SSs (type IV secretion systems) proteins necessary for the conveyance of CagA into gastric host cells [82,83]. Recently, it was shown that *H. pylori*, which acts as a causative agent of severe gastric disease, is a significant attraction center for research [82]. A study reports the functional implication of calgranulin C (S100A12) in regulating *H. pylori* growth [84]. For instance, it has been elucidated that, in a dose-dependent manner, calprotectin efficiently alters numerous activities of *H. pylori*, such as the modification of lipids [85], a structural component of the outer membrane, and the slowing down of the *cag*-Type IV secretion process mainly responsible for pathogenesis. Moreover, S100A2 can bind Zn and limit the availability of Zn micronutrients required for the growth or proliferation of *H. pylori* to provide nutritional immunity against it [86].

### 2.3. Role of S100 Protein in Various Immunological Process

#### 2.3.1. S100 Protein Could Be the Prognostic Marker for COVID-19

The epidemic of COVID-19 has become the greatest global public health disaster worldwide. As of 10 July 2022, more than 555 million infections and 6.35 million confirmed deaths had been documented globally. This below-included web link will allow you to determine the current update number (https://www.google.com/search?client=firefox-bd&q=world+covid+casis#colocmid=/m/02j71&coasync=0) (Last visited on 10 July 2022). Recent publications have explored potential clinical interventions, including the use of the S100 gene family as a prognostic marker based on omics data from COVID-19 virus–host interactions and immune responses. The S100 family of genes (S100A6, S100B, S100A8, S100A9, S100A12, and S100P) was identified as a key category of host factors that appeared at the end of the meta-analysis, as well as being validated in the COVID-19 cohort. Multiple genes from the S100 family, such as S100A8, S100A9, S100A6, S100A11, and S100P, as well as a few other genes, such as ASS1, neutrophil defensin alpha 3 (DEFA3), and SERPINB3, were significantly upregulated in patients with positive symptoms. This indicates that they may have diagnostic and prognostic value which is independent of age and gender [87]. However, multiple investigations have assessed transcriptional and proteomic changes in moderate, severe, and fatal COVID-19 cases to find diagnostic and prognostic serum signs [88,89] (Figure 5).

Serum S100A8/A9 heterodimer (calprotectin) levels have been linked to disease severity and an excess of cytokines [90,91]. Patients with fatal COVID-19 infections had overexpression of S100A12, S100A8, S100A9, and S100P in transcriptomic analyses of lung tissue [92]. All of the above-mentioned S100s (excluding S100A12) displayed significant sensitivity as predictive markers of symptomatic COVID-19, according to the ROC curve analysis of the Positive Asymptomatic and Positive Symptomatic group gene expression data [87]. Furthermore, S100B levels were also found to be significantly higher in mild and severe disease cohorts than in healthy controls [93,94]. However, some previous studies showed a correlation between S100B and pulmonary inflammation, with S100B being upregulated in bronchiolar epithelial cells and airway dendritic cells [95,96]. However, the source of increased serum S100B in COVID-19 patients has yet to be identified.

In addition, a recent study demonstrated that S100A4, S100A9, and S100A10 have a role in the inflammatory conditions, as well as the severity, of COVID-19 patients, and have the ability to influence the prognosis of the severe form of the disease [97]. This study shows a link between S100A4, the S100A8/A9 heterodimer, and S100A10 and LDH levels, suggesting these molecules contribute to acute lung injury and ARDS (acute respiratory distress syndrome) [98]. According to some studies, S100A4, S100A9, and S100A10 expression is proportional to neutrophil/lymphocyte ratio, and may reduce peripheral blood lymphocytes in COVID-19 patients [97]. Chen et al. relate blood concentrations of S100A8/A9 heterodimer with concentrations of a variety of pro-inflammatory cytokines, most notably IL-8, MCP-3, MCP-1, IL-1ra, CTACK, β-NGF, IL-7, IL-10, RANTES, G-CSF, IL-1α, and IL-17A [90]. However, Bagheri et al. demonstrated a significant association between the expression of S100A4, S100A9, and S100A10 and inflammatory indices (CRP (C-reactive protein), ESR (erythrocyte sedimentation rate)), and elevated leukocytosis in COVID-19 patients [97]. Based on these results, the S100 family may be able to control cytokine release syndrome and get more monocytes and neutrophils to the target sites in COVID-19 patients.

When researchers attempt to determine if S100A8 levels rise in other viral infections, such as encephalomyocarditis virus (EMCV), herpes simplex virus 1 (HSV-1), and influenza A virus (IAV), the authors discovered that its levels are elicited solely by the COVID-19 virus. In addition, the author also examined an increase of S100A8 in MHV (Mouse hepatitis virus). Keeping together, the coronaviruses, COVID-19 and MHV, elicited a nearly homogeneous immune response. This indicates that coronaviruses, but not other viruses, induce abnormal expression of S100A8 [99]. It is difficult to explain how S100A8 regulates the pathogenesis of COVID-19 because S100A8 plays a critical function in immunological responses. As of right now, it is unclear if S100 protein regulates COVID-19 infection in a positive or negative way.

Under normal physiological settings, neutrophils and myeloid-derived dendritic cells retain enormous amounts of S100A8 and S100A9, whereas monocytes express modest quantities of S100A8 and S100A9 constitutively [100,101]. In the lungs of rhesus macaques infected with COVID-19 virus, markers for monocytes and natural killer cells were marginally elevated, T cells were unaffected, and B cells were considerably downregulated [99]. Recently, it has been studied how COVID-19 infection activates anti-bacterial responses, by analyzing the differential expression of genes before and after infection. In addition, they also discovered that S100A8 was the most strongly upregulated gene of all known alarmins [100].

In mice infected with coronavirus, neutrophils were deformed. The majority of neutrophils in mice infected with COVID-19 and MHV were CD45 + CD11b + Ly6G^varying^, when compared to neutrophils in the control group, which were CD45 + CD11b + Ly6G^high^ [100]. This indicates that a population of dysplastic aberrant neutrophils was produced by the coronavirus infection, which could lead to deregulation of the innate immune system. To determine if S100A8, which is a major cytoplasmic protein of neutrophils, influences neutrophil activity, paquinimod, an inhibitor of S100A8/A9 heterodimer binding to TLR4, was used. Compared to the coronavirus infection group, the majority of neutrophils in mice treated with Paquinimod reverted to normal CD45 + CD11b + Ly6G^high^ levels, thereby rescuing the mice from a fatal outcome due to coronavirus infection. In addition, other recent studies also found that these aberrant neutrophils exhibited obvious immature characteristics [100,101,102,103,104,105]. Studies indicate that S100A8 can be used as a prognostic marker for COVID-19-positive patients and could be the most effective treatment target for COVID-19 by blocking the S100A8/A9 heterodimer binding to the TLR receptor. However, additional studies are necessary to clinically demonstrate the most effective therapy target against COVID-19.

#### 2.3.2. Functional Contacts of Nerves with Immune Cells through S100 Protein 

In normal conditions, S100 is known for its function in neurite growth and supports the viability of neurons [15]. Recently, an altered concentration of S100 induces proinflammatory cytokines, such as IL-1β, TNF-α, and NO synthetase (stress-inducing enzyme). Moreover, S100-dependent induction of NO formation in astrocytes leads to neuronal death [106]. Glaucoma is an eye disorder associated with vision loss and blindness caused by damage of the optic nerves and the gradual death of RGCs (Retinal Ganglion Cells) with intraocular pressure (high eye pressure) characteristics. The latest research output suggests the significant contribution of immunological function to multifactor mediated glaucoma through the S100 protein. The study used an autoimmune glaucoma model to explain the immune system-related process in the nervous system [107]. Exogenous insertion of S100B (used as an ocular antigen) in the glaucoma model caused a loss of RGCs (Retinal Ganglion Cells) and degeneration of the optic nerve after 28 days of the window, without intraocular pressure. They also detected a high number of microglial cells (macrophage cells of the CNS (Central Nervous System) and autoantibodies in RGCs and optic nerves after the treatment of S100B [107]. TLR-4 plays a role in neuronal cell death in the CNS, microglial cell life in optic nerves and RGCs, and complement-pathway protein secretion through retinal microglial cells during optic nerve injury disease, providing insight into the immune system’s functional intervention through S100B activation. The induction of TLR-4/NF-κB pathway proteins by S100B enhances neuroinflammation by activating the innate immune response (complement activation). In addition, S100B-induced NF-κB in microglial cells govern cells’ chemotaxis movement toward the injury site via β-integrin CD11a expression. As a result, it can be concluded that S100B-mediated activation of NF-κB and complement pathways plays a vital role in the pathogenesis of glaucoma [107]. 

Therefore, exogenous insertion of S100B in vitreous humor confirms the direct/indirect function implication of S100B protein activation of the above-mentioned late systemic immune response during glaucoma, and begins from the degeneration of both retinal ganglion optic nerves, leading to the brokerage of the blood–retinal barrier (BRB). Intact blood–retinal barriers usually regulate the immigration of immune cells from the choroid to the sub-retinal space. Altered or compromised integrity of the BRB increases ocular hypertension and accumulation of B-cells in the retina. Hence, compromised porous BRB further facilitates immune response strengthening of the degeneration of retinal ganglion cells and nerves in the eyes. It is known that apoptosis is an earlier phenomenon, that occurs during the degeneration of the ganglion and optic nerve. A high level of S100B activates the caspase-mediated cell death cascade during degeneration by increasing the level of active caspase 3 [108].

Cross-communication between the nervous and immune systems is critical for immune system regulation, and is mainly regulated by the HPA (Hypothalamic–Pituitary–Adrenal) axis and the SNS (Sympathetic Nervous System) [109]. The latest hardwired neural pathway elucidates the contact connection between sympathetic nerves and immune cells in lymphoid tissue. Moreover, S100-positive cells in cervical lymph nodes are directly targeted by nerve fibers from the superior cervical ganglion. Furthermore, the transmission of a signal from the CNS to immune cells is mediated by the expression of neurotransmitters, such as neuropeptide Y, norepinephrine, and vasoactive intestinal polypeptide, by postganglionic nerve fibers of the extremity, which innervate S100+ cells to induce a further immune response in lymphatic tissue. Thus, it concludes that the cross-talk communicable approach between the nervous system and the immune system plays a crucial role in transmitting messages or signals from central nervous system nerve cells to targeted S100 positive immune cells in lymphatic organs [110].

In nervous system disorders, such as early-onset Alzheimer’s disease (AD) and bacterial meningitis, a member of the S100 protein family has been identified as a potential candidate. Several studies have shown the existence of S100 proteins within or near protein inclusions, including those within β-amyloid (Aβ) aggregation and others in astrocytes and microglial cells located near the A aggregation, implying that this protein plays a significant role in AD [111,112,113,114]. Excess Zn^+2^ ions induce neurotoxicity in nerves, perhaps by aiding in the deposition of β-amyloid (Aβ), leading to plaque formation, which is the pathogenic systematic hallmark molecular pattern for AD brain. It has been found that astrocyte-originated S100A6 [111] and S100B [112] proteins effectively regulate Zn^+2^ elevation, and subsequently hamper Zn^+2^-mediated plaque formation (Aβ aggregation) by chelating the zinc ions to inhibit. However, astrocyte and microglial cells enhance the production and release of numerous S100 proteins around the plaque inclusion to contribute to several misregulated molecular processes during AD. For instance, S100A1, S100B, and S100A6 involve NETosis, disassembly of the cytoskeleton, and Tau phosphorylation. Contrarily, S100B and S100A9 contribute to neurofibrillary tangles. Several members are involved in amyloid precursor protein (APP) processing, which generates Aβ peptide through proteolytic digestion of type I transmembrane protein (APP). S100A9 controls the activity and expression of β-/γ-secretase (an enzyme responsible for proteolysis of APP [115]. S100B and S100A1 govern the level of APP. S100A8, S100A7, S100B, and S100A9 influence Aβ levels. Furthermore, zinc homeostasis is maintained through the zinc buffering activity of S100B and S100A6. In addition, S100A1, S100B, and S1009 potentiate engagement of the Aβ peptide and inhibit aggregation [114].

Bacterial meningitis is a nervous system-associated inflammatory disease characterized by the severe inflammatory response of meningeal cells (dura mater, arachnoid mater, pia mater, and the subarachnoid space) to the blood–brain barrier of the brain. Astrocytes are prime cells for structural support and management of the blood–brain barrier. Therefore, it they play a significant role in inflammation, neurodegeneration apoptosis, and bacterial and viral strikes. In addition, these cells participate in the innate immune response to combat bacterial meningitis or viral infection by secreting various AMPs, such as cathelicidin, defensins, and S100A15, during an inflammatory situation. Moreover, meningeal cells, or glial cells, also initiate the release of multiple inflammation modulators. Accumulation of S100A7 during the onset of AD, in early mild cognitive impairment, and inflammation, as AMPS, during bacterial/viral infection in CNF has been reported [116].

Although there was little evidence regarding the existence of S100A4 in the CNS, it was recently discovered that S100A4 is highly expressed in activated microglial cells of the CNS in mice, and that niclosamide inhibits its transcription. Additionally, it has been shown that amyotrophic lateral sclerosis patients’ astrocytes, microglial cells, and fibroblasts have increased levels of S100A4. This indicates that S100A4 has a functional role in microglial reactivity, contributing to the regulation of neuroinflammation [117].

#### 2.3.3. Involvement of S100 Protein in Autoimmune Disease or Immune System-Related Disease

An autoimmune disorder is a clinical disease condition where host defense mechanisms initiate recognizing their own body units (such as cells or tissue) as the foreign invader’s entity and damage their own body tissue through autoimmunity. One of the primary defense mechanisms of the systemic immune system, known as sterile inflammation, acts as a driving factor for various chronic inflammatory diseases, such as autoimmune disease [118,119], atherosclerosis [120], psoriasis [58], intestinal bowel disease [121,122], sepsis [51], rheumatoid arthritis [123], glaucoma [107], and liver disease [124] (Figure 6). 


a.Rheumatoid arthritis (RA)


RA is an autoimmune disorder that includes 200 distinct diseases. It is characterized by inflammation in the synovial tissue of the joints and progresses to cartilage and bone loss. Endogenous and exogenous proinflammatory factors, or alarmins (DAMPs), induce inflammation by interacting with PRRs, such as RAGE, TLR-4, GPCR, and EMMPRIN (also known as basigin or CD147), which is a well-known process in rheumatoid arthritis or synovial tissue damage [125]. Some S100 protein members are abundantly expressed in rheumatic diseases. The S100A8/A9 heterodimer, S100A4, S100A11, S100A12, and S100B are well known in rheumatic diseases. Serum S100A4 triggers TLR4 receptor activation to induce expression of IL-1β, IL-6, and TNF-α by stimulation of peripheral blood mononuclear cells, and leads to an expression release of MMPs in synovial fibroblasts. Similarly, in RA disease, S100A8 and/or S100A9 activate synovial macrophages and enhance TNF-α expression, IL-6, and IL-1β in monocytes, and induce MMPs to mediate chondrocyte dependent cartilage destruction [126].


b.Osteoarthritis (OA)


OA is another example of an autoimmune RA related disease, where S100 protein members play a critical role in disease pathogenesis through RAGE and TLR-4 receptor engagement dependent manner. In OA, expression of the S100A8/A9 heterodimer has been reported in the synovium and cartilage of joints. Elevation of S100A8/A9 during osteophyte formation in humans has been demonstrated by elevated S100A8/A9 plasma levels in people with early symptomatic OA. Using S100A9-KO mice as a model for OA, the author discovered that S100A8 and S100A9 are required for the formation of large osteophytes at both the bone margins and in ligaments. Previously, it has been shown that cartilage damage is reduced in S100A9-monomer-KO mice during OA [127]. However, a recent study found that S100A8 and S100A9, which are important products of activated macrophages during synovial activation in OA, may increase osteophyte size in experimental OA with synovial inflammation. The S100A8/A9 heterodimer has the ability to upregulate and activate MMPs, which aid in cartilage matrix remodelling and allow osteophytes to grow in size [128]. The S100A8/A9 heterodimer may, thus, be a useful biomarker for predicting cartilage damage and osteophyte progression in human OA. S100A8 and S100A9 enhance interleukins expression from immune cells and formation of osteophyte. In addition, S100A8 is connected to pain generation in OA. S100A10 contributes to MAPK and NF-κB-mediated production of inflammatory cytokines in chondrocytes. S100A4, in OA, similar to RA, enhances the expression of MMP13 through stimulating the activation of MAPK, PYK2, and NF-κB in chondrocytes. Similarly, S100B enhances the expression of MMP-13, mediated by ERK and NF-κB in chondrocytes. Increased expression of S100A12 found in articular cartilage during OA, as S100A4 and S100B, increases expression of MMP-13 and VEGF in MAPK, p38, and NF-κB manner [41]. Immune cells related to the elevated level of the S100A8/A9 heterodimer and S100A12 are a significant biomarker of treatment response in juvenile idiopathic arthritis in adults [129,130] and children upon anti-TNF-α therapy [131]. 


c.Osteoporosis


Recently, De Martinis et al. compiled information in a review related to the functional contribution of alarmins in osteoporosis and arthritis [46]. Osteoporosis is a progressive inflammatory condition characterized by decreased bone mass and the destruction of bone microarchitecture, resulting in a loss of physical strength of the body’s skeleton and an increased risk of bone fracture caused by RA [132]. The alarmin S100 protein is released by leukocytes during inflammation and interacts with extracellular receptors; for example, S100A12 and S100B bind to RAGE, while S100A8/S100A9 bind to TLR-4. Thus, Alarmin S100A8 is a potentiate member that stimulates osteoclast cells by interacting with TLR-4 to improve bone structure remodelling by maintenance and repair. Alarmin S100A9, on the other hand, increases RAGE expression and promotes cytokine release in bone synthesizing, or forming osteoblast cells. Furthermore, S100A9-treated osteoblasts promote the differentiation and activity of osteoclast progenitor cells [46]. Likewise, S100A16 plays a role in osteoblast differentiation and negative interference with osteogenesis by promoting adipogenesis through upregulation of PPAR (Peroxisome Proliferator-Activated Receptor-ƴ) and downregulation of RUNX2 (Runt-Related Transcription Factor-2) transcriptional expression [133]. 


d.Psoriasis


Psoriasis is another well-known autoimmune disorder characterized by chronic inflammation with inflamed, red, and scaling skin areas caused by misdirected T-cells, dendritic cells, and inflammatory cytokines that attack the skin and induce uncontrolled keratinocyte proliferation. One of the S100 protein members is known as psoriasin (S100A7) because it is seen in psoriasis and other skin diseases. However, there is extensive information in the literature about the role of S100A7 in psoriasis. Small subsets of S100 (for example, S100A7, S100A8, S100A9, and S100A12) have been shown to be upregulated in psoriasis skin lesions, whereas transcriptomics and ELISA-based approaches indicate that S100A12 is strongly correlated with a functional disease condition [134,135] (Figure 6). S100A4 [136] and S100B [137] have also been implicated in the pathogenesis of psoriasis. In support of the preceding finding, new research has revealed significantly elevated expression of alarmins, such as IL-33, HMGB1, S100A7, and S100A12, in serum, implying a role for these alarmins in the immunopathology of psoriasis conditions [138]. Regardless of infected cells, many autoimmune and inflammatory diseases stimulate the release of endogenous alarmin factors into the extracellular environment, where they interact with corresponding receptors on immune cells to enhance innate immune response, cell differentiation or death, and inflammation regulatory pathways. High throughput analysis revealed that the role of major alarmins, such as S100 proteins (S100A6 and S100A9), HMGB1, and HSPs (heat shock protein), is required for the establishment and exacerbation of inflammation, hyperglycemia, cancer, and atherosclerosis [139]. 


e.Atherosclerosis


Atherosclerosis is a chronic inflammatory disease caused by plaque formation in an artery’s intima [140]. Initiation of atherosclerotic plaque formation involves oxidized LDL (oxLDL), dendritic cells, macrophages, foam cells (FCs), and monocytes. LDL molecules accumulate in the tunica, causing dysregulation and dysfunction of endothelial and smooth muscle cells (SMCs), resulting in proinflammatory cytokine secretion. 

Monocytes in the bloodstream sense the cytokine and move to sub-endothelial space, attracting the atherosclerotic plaque. These cells differentiate into macrophages with scavenger receptors, which engulf oxLDL. Active macrophages become foam cells after ingesting oxLDL [141]. T lymphocytes enter the tunica and control the innate immune response later. Smooth muscle cells release necrotic matrices and ECM proteins (collagen and elastin) to form the fibrous cap, which covers the lipid core, oxLDL, and necrotic cells to stabilize the plaque [142,143,144] (Figure 7). During the formation of arterial plaques, calgranulins begin to engage their corresponding receptors, such as RAGE, TLR-4, and CD36, thereby contributing to immune-cell response. These members are specifically produced by monocytes, vascular endothelial cells, and SMCs in response to oxidative stress caused by atherosclerosis. These secretory S100 proteins binds to the TLR-4 receptor, activating downstream signaling and increasing NF-κB and ROS production. It begins by increasing pro-inflammatory activity in a variety of cells in blood vessel extremities, including endothelium, leukocytes, and SMCs (Figure 7). Inhibiting S100 protein-mediated inflammation in arterial plaque formation could be a promising atherosclerosis treatment [145,146]. 


f.Inflammatory bowel disease (IBD)


IBD is an immune-related disease triggered by persistent inflammation in the digestive tract, resulting in digestive problems. There are two subtypes of IBD: ulcerative colitis (UC) and Crohn’s disease (CD). UC is characterized by inflammation of the colon or large intestine, while CD affects various parts of the digestive tract from the mouth to the anus, such as the small intestine’s final segment before it enters the colon [122]. The S100A8/A9 heterodimer (calprotectin), which regulates the inflammatory process, is associated with chronic inflammatory gut disease, or IBD (Figure 6). It has been noticed that trace residues of calprotectin in the fecal matter of IBD patients suggest calprotectin as a non-invasive marker for IBD [147,148,149]. Active neutrophils containing ~60% calprotectin content in the cytosol initiate travel to the intestinal mucosa from the circulatory system during active intestinal inflammation. Any damage due to inflammation of the inner lining of the intestinal mucosal membrane leads to leakage of neutrophils, resulting in the release of calprotectin from neutrophils into the lumen, and, subsequently, into the feces [121,149]. 


g.Chronic rhinosinusitis (CRS)


S100 members have recently addressed another local and systemic inflammatory condition in the nose, CRS. CRS patients have decreased levels of S100A7 and S100A8/A9, according to research by Kim et al. [150]. During CRS, psoriasin and calgranulins (S100A8, S100A9, and S100A12), which are known for their chemo-attractive properties to immune cells, increased pro-inflammation and triggered proliferation via TLR-4 and RAGE [151,152]. Moreover, recent research by Boruk et al., confirmed that S100A9, MMP3, MMP7, MMP11, MMP25, MMP28, and CTSK protein levels are elevated in CRS nasal tissues. The proliferation of nasal epithelial cells is induced by S100A9. These findings suggest that MMP3 is sensitive to S100A9 signaling, and that both molecules contribute to nasal epithelial cell proliferation [153]. More research is necessary to confirm whether S100A9 directly contributes to CRS progression.

#### 2.3.4. The Immune System Regulates the Expression of S100 Protein during Pregnancy 

The mother’s immune system plays a significant role during the successful progression of a healthy pregnancy, specifically in the establishment, maintenance, and completion of the pregnancy. Several immune cells and factors play an essential role in the formation and function of the placenta, which serves as a temporary physical connection between the embryo and the mother. Successful establishment of pregnancy necessitates a delicate balance between effector immune cells such as Th1 (T helper 1) and Th2 (T helper 2), as well as pro-inflammatory and anti-inflammatory factors. In brief, immune cells of the immune system begin to accumulate in the endometrium during decidualization, and perform various functions at the maternal–embryo interface, suggesting that the immune system plays a critical role, specifically during embryo implantation and placental connection development, as well as during immunity generation against pathogenic disease [154,155]. During a normal pregnancy, for example, modified endometrium or decidua accommodate an abundance of immune cells, including 70 percent of uterine natural (uNK) cells, 20–25 percent of macrophages, 1.7 percent of uterine dendritic cells (uDC), and 3–10 percent of regulatory T-cells. Furthermore, it has been demonstrated that, during the first trimester, uDC, macrophage, and uNK cells penetrate the decidua and begin to gather near the overwhelmed trophoblast cells, indicating that uNK cells pre-request and necessitate to trophoblast cells invading the endometrium. In addition, uDC plays a critical role during blastocyst implantation and decasualization, and influences angiogenic response by hampering blood vessel maturation [156,157].

Both immune and non-immune cells can express and release the S100 protein. Calgranulins, for example, are primarily released by granulocytes, the early stage of macrophages and monocytes (myeloid cells) [158]. In addition, it is known that uNKs, macrophages, T-regs, and neutrophils are responsible for regulating and maintaining immune responses for a successful pregnancy. As a result, any change in the inflammatory and immunomodulatory pathways could result in increased expression and release of S100 protein via non-immune cells. Furthermore, S100 proteins, which includes S100A11, S100A10, S100A8, S100A9, S100P, S100A6, S100G, and S100B, play an important role in pregnancy progression from non-immune cells. 

S10011 was found to be upregulated during a successful pregnancy, and it plays a critical role in embryo implantation and endometrium receptivity via the EGF-AKT pathway, as well as increasing the TH2/TH1 ratio. S100A10, which is released by endometrium stromal cells during the mid-secretory phase, also increases endometrium receptivity and immune tolerance by inducing apoptosis via annexin 2 and regulating prolactin secretion. S100A8 is a protein found in the uterine fluid, embryo, and maternal vasculature that regulates preimplantation, to prevent embryo rejection, by regulating the PIF molecular pathway and post-implantation maternal angiogenesis regulation. Similarly, S100P is found at a higher level during the receptive phase of the endometrium and is released by endometrial stromal/epithelial cells, the placenta, and the trophoblast. It regulates endometrial receptivity through a molecular pathway involving RAGE, MAPK, placental ERK, and trophoblast NF-kB. After implantation, S100A6 (calcyclin) is found in higher concentrations in the decidua to induce placental lactogen (human chorionic somatomammotroph (CSH) or human chorionic lactogen) secretion from the placenta and trophoblast. It is also secreted by the uterus’ NK cells during pregnancy. S100G expression is low during embryo implantation via epithelium luminal cells and glandular epithelium, and aids in trophoblast invasion by inducing apoptosis and increasing free calcium. To control calcium channel function, luminal and glandular epithelial cells secrete more S100B during the luteal phase than during the follicular phase during embryo implantation [159,160]. The S100 protein is primarily involved in embryo adhesion/implantation, endometrial receptivity, immune tolerance, prolactin secretion, and endometrial epithelial cell apoptosis, implying conclusively that it is a hallmark marker for the onset of decasualization [159]. 

In brief, during normal pregnancy, active homo- and heterodimer formation and expression of the S100 protein are necessary for proper embryo adhesion, immune tolerance, and decidualization. In addition, by increasing IL-10 and decreasing other cytokines (such as TNF-α, INF-ƴ, IL-2, and IL-12), myeloid cells, such as uNK, T-reg, macrophages, and neutrophils, maintain the delicate balance between Th1/Th2 or pro/anti-inflammatory ratios to establish a satisfactory innate and adaptive response (Figure 8). Therefore, any alteration due to pathological conditions in cytokine release and count of myeloid cells due to any circumstances could lead to a disturbance in Th1/Th2 or pro/anti-inflammatory ratios, resulting in an alteration in the expression of S100 protein by immune non-immune cells. This results in altered S100 protein expression, which causes pregnancy-related complications, such as embryo implantation failure, immune tolerance dysregulation, and improper decidualization or decidua formation [160,161,162,163]. 

According to a recent report, S100A8 levels in the peripheral blood are elevated in preeclampsia during pregnancy. In addition, differences in inflammatory factor expression have been discovered as preeclampsia progresses. For example, late-onset severe preeclampsia (LS-PE) has higher transcriptional expression of pro-inflammatory cytokines and lower levels of anti-inflammatory cytokines than early-onset severe preeclampsia (ES-PE). Furthermore, S100A8 expression strongly correlates with IL-12, IL-6, and TNF-α and negatively correlates with IL-10, indicating that inflammatory cytokines and S100 protein interact during preeclampsia [164]. 

## 3. Conclusions

S100 proteins have been demonstrated to be required by many species for their defense systems. In addition, S100 isoforms serve as an alarm, antimicrobial peptide, pro-inflammatory stimulant, chemoattractant, and metal scavenger during an innate immune response. Thus, they are critical in the treatment of autoimmune diseases. In this review, we explored the several roles of S100 in the immune system and its related processes. First, the S100-family molecular-binding factors control immunological processes. Second, increasing data suggest that the S100 protein acts as an inflammatory regulator. On the other hand, phagocytic cells that are in close proximity to inflamed tissues produce S100A8, S100A9, and S100A12. Third, extracellular S100 regulates cell death, differentiation, proliferation, and chemoattraction (migration) in a variety of cell types, including immunological, epithelial, stromal, fibroblast, neuron, endothelial, and smooth muscle cells. Forth, S100 proteins possess antimicrobial activity. Finally, correlation between S100A6, S100B, S100A8, S100A9, S100A12, and S100P and COVID-19 pathogenesis is discussed. In addition, an increase surge in S100A8 and S100B is associated with mild to severe COVID-19 pathogenesis. Increased levels of both proteins could be used as a biomarker for the prognosis of COVID-19 patients. We have shown, in detail, how S100 proteins work in neutrophils, macrophages, inflammation, ageing, pregnancy, and other autoimmune diseases.

## Figures and Tables

**Figure 1 cells-11-02274-f001:**
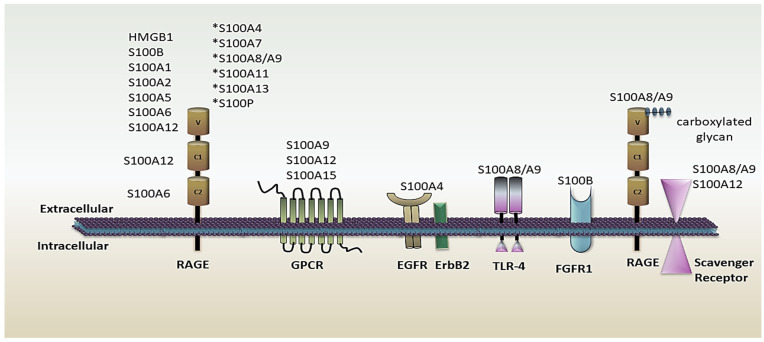
S100 isoforms interact with a range of cell surface receptors, such as GPCR, RAGE, TLR-4, and proteoglycans heparin sulphate and N glycan scavenger receptors. GPCR, G protein-coupled receptors; RAGE, Receptor for Advanced Glycation End-Products; TLR-4, Toll-Like Receptor-4. * S100A4, S100A7, S100A8/A9, S100A11, S100A13, and S100P interact with and/or activate RAGE, but the exact domain is unknown.

**Figure 2 cells-11-02274-f002:**
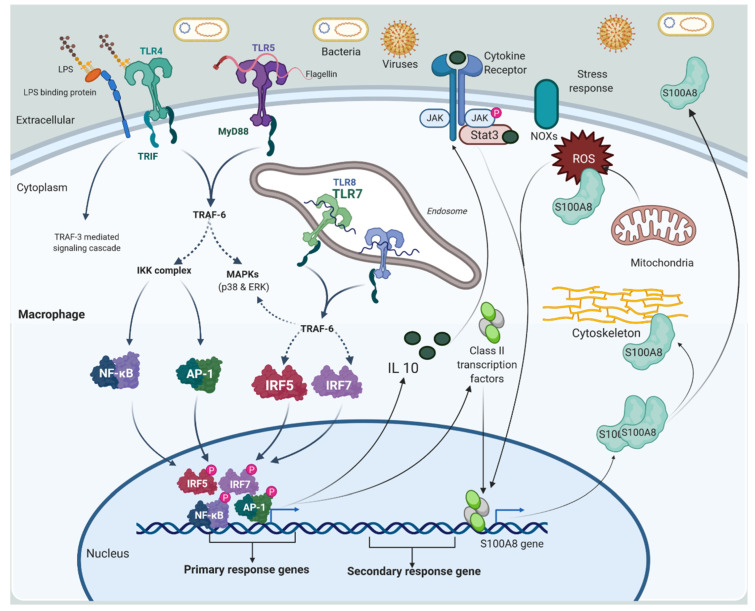
The activation mechanism of macrophages is depicted. Bacterial LPS and endotoxins cause phagocytic macrophages to activate. This activates the TLR-4 receptor on macrophage surfaces, which triggers S100A8 expression. TLR-4 activation enhances the signal via the downstream signaling cascade, activating NF-κB, AP-1, and IRF-3 transcription factors via non-endosomal and endosomal TLR-4 pathways. These transcriptional regulatory factors regulate primary response genes, IL-10 (an anti-inflammatory cytokine), and class II transcriptional factors, such as C/EBPs, AP-1, and Stat-3. In addition, the expression of S100A8 as a secondary response gene, or late gene, should be raised. IL-10 promotes the expression of S100A8 in macrophages. S100A8 works as an oxidant scavenger in a heterodimer with S100A9, interacting with cytoskeletal proteins for cytoskeleton reorganization and secreting, into the extracellular matrix, via non-classical secretory pathways, its extracellular activity. LPS (Lipopolysaccharides). Created with BioRender.com.

**Figure 3 cells-11-02274-f003:**
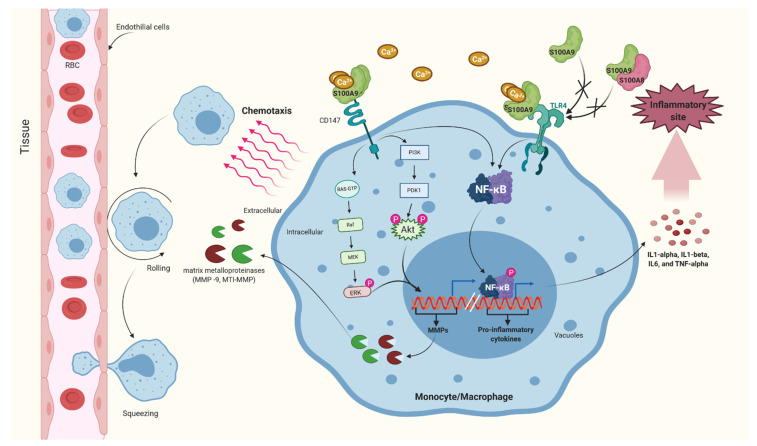
The image depicts the S100 isoform, S100A9, which stimulates the release of pro-inflammatory cytokines by binding to the TLR-4 receptor, which activates the NF-κB transcription factor, resulting in the expression of pro-inflammatory response genes in monocytes. Created with BioRender.com.

**Figure 4 cells-11-02274-f004:**
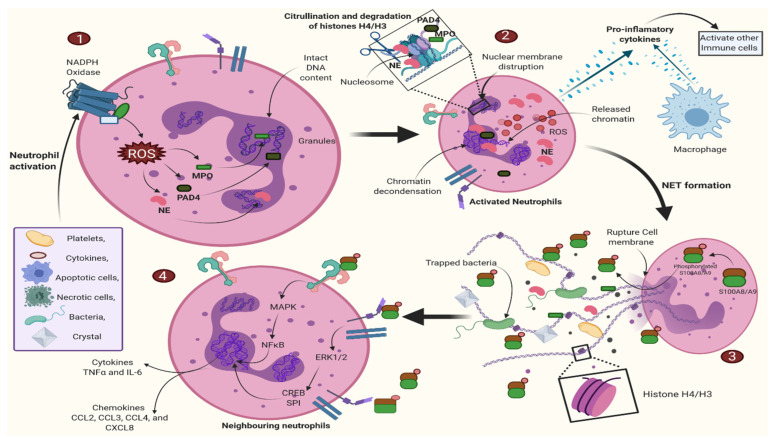
S100A8/PhosphoA9 induces a pro-inflammatory response. Neutrophils stimulated by various stimuli (PMA, MSU, *Aspergillus fumigates*, or *Aspergillus nidulans*) release NETs via a pathway involving NADPH oxidase, PAD4, NE, and MPO. During NET formation, the phosphorylated S100A8/A9 heterodimer is released into the extracellular space. S100A8/PhosphoA9 can then activate neutrophils in the surrounding area, causing them to release cytokines (TNF-α and IL-6) and chemokines (CCL2, CCL3, CCL4, and CXCL8). TLR4 signaling pathways are primarily responsible for this release, while additional receptors (such as RAGE) are involved in S100A8/PhosphoA9-mediated CCL2 secretion. As a result, S100A8/PhosphoA9 produced by neutrophils is implicated in amplifying the inflammatory process, and may be a defining feature of inflammatory disorders. Here, MPO stands for myeloperoxidase; MSU stands for monosodium urate monohydrate; NE stands for neutrophil elastase; PAD4 stands for peptidyl arginine deiminase; PMA stands for phorbol 12-myristoyl 13-acetate; NET stands for a neutrophil extracellular trap; NADPH oxidase stands for nicotinamide adenine dinucleotide [35]. Created with BioRender.com.

**Figure 5 cells-11-02274-f005:**
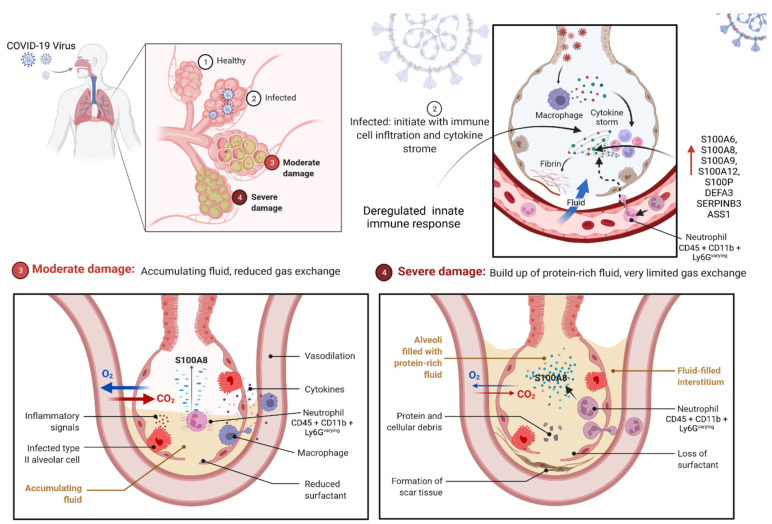
This diagram depicts the progression of a COVID-19 infection from the healthy to fatal stage. COVID-19 disrupts the immune response by inducing a cytokine storm and S100A8 overexpression. Infection with COVID-19 induces the formation of abnormal neutrophils with variable CD45 + CD11b + Ly6G markers, causing these cells to behave abnormally. Created with BioRender.com.

**Figure 6 cells-11-02274-f006:**
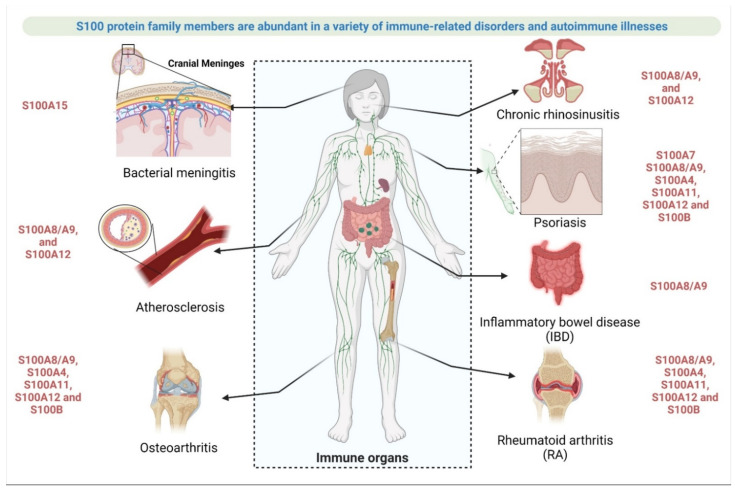
Detailing the role of an S100 protein family member in a variety of immune-related disorders and autoimmune illnesses, in which S100 proteins are abundantly expressed. Created with BioRender.com.

**Figure 7 cells-11-02274-f007:**
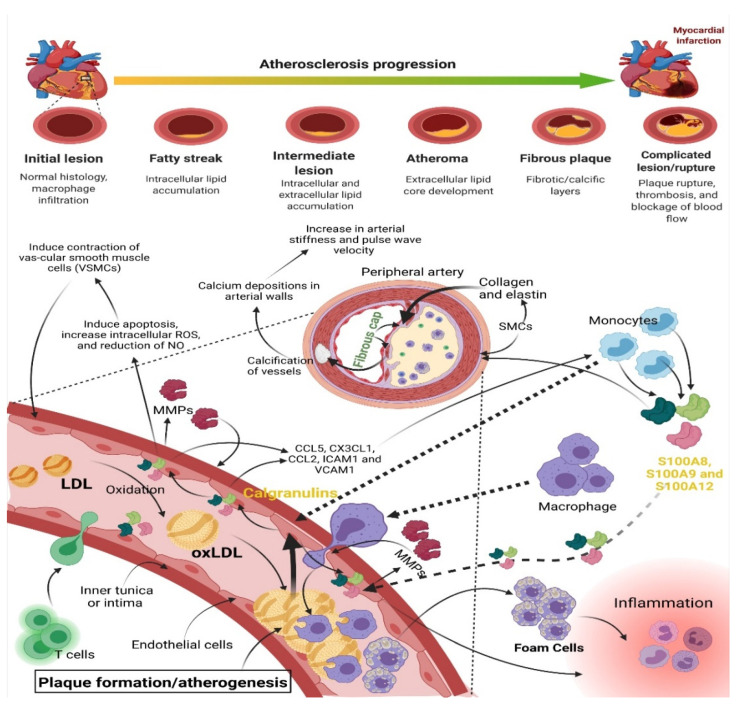
An illustration of the progression of atherosclerosis, a chronic inflammatory disease. The accumulation of oxidized LDL (oxLDL) triggers the recruitment of macrophages, which consume oxLDL and transform into foam cells. This mechanism causes endothelial and SMC dysregulation and malfunction, leading to pro-inflammatory cytokines, such as CCL5, CX3CL1, CCL2, ICAM1, and VCAM1, into the bloodstream. In addition, this results in the recruitment of even more immune cells, including monocyte and T-cells. Activated monocytes secrete calgranulins, which activate TLR-4 and RAGE receptors. In addition, the S100A8/A9 heterodimer and A12 activate SMC and endothelial cells, and activated endothelial cells induce apoptosis. In addition, this initiates and increases intracellular ROS while causing a decrease in NO and the release of MMPs, which ultimately causes contraction of VSMCs and inflammation. SMC, on the other hand, secretes collagen and elastin, which form a fibrous cap. NO, nitrous oxide; SMC, smooth muscle cell; RAGE, Receptor For Advanced Glycation End-Products; TLR-4, Toll-Like Receptor-4; ROS, reactive oxygen species; VSMCs, vascular smooth muscle cells; MMPs, matrix metalloproteinases. CCL5, CX3CL1, CCL2; chemotactic cytokine or chemokine. ICAM1, Intercellular Adhesion Molecule 1. Created with BioRen-der.com.

**Figure 8 cells-11-02274-f008:**
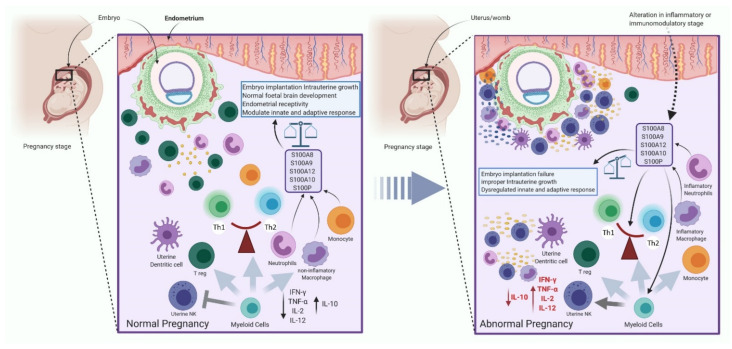
The interaction of S100 proteins with immune cells to control different characteristic stages of pregnancy is depicted in this diagram. TNF-α, tumor necrosis factor-alpha; uNK, natural uterine killer; IFN-γ, interferon-gamma; IL, interleukin; TH, T helper. Created with BioRender.com.

## Data Availability

The study did not report any data.

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
