# Peer review of "Multifunctional Role of S100 Protein Family in the Immune System: An Update"

_cells, 2022, doi:10.3390/cells11152274_

Round 1

Reviewer 1 Report

In the review by Parul Singh and Syed Azmal Ali entitled “Multifunctional role of S100 protein family in the immune system: an update” the authors covered the multifaced implications of S100 protein family members in the immune system, particularly the role of S100 proteins as alarmins, antimicrobial peptides and metal-ion scavengers during innate immune responses. Further, the manuscript describes the implication of multiple S100 family proteins in the pathophysiology of COVID-19 and roles of S100A8 and S100A9 as potential prognostic markers and promising therapeutic targets. The authors also have managed to summarize a number of important features of known roles of S100 proteins in autoimmune and immune system-associated diseases. Overall, this is a useful review that aims to update what is known about the role of the S100 protein family in immune system. However, the review is somewhat confusing due to the large number of acronyms and abbreviations. A complete list of them is needed. In addition, a list of all alternative names, such as calgranulin, calprotectin, and calcyclin, would help to understand which S100 family proteins function in the context of the immune system. The following are other concerns, comments, and questions.

Line 35 and following: a doubly charged cation should be indicated as 2+ instead of +2.

Figure 1. It would be helpful to explain the domain organisation of the receptor and illustrate the binding sites of the S100 proteins to help understanding. In figure legend (line 55) and text (line 61), “G-Coupled Receptors” replace by: G protein-coupled receptors.

Line 120 and following: it is confusing whether IL-10 affects the expression of S100A8 (lines 120-121) or not (lines 128-129) in macrophages.

Line 159 and following: “S100A9/A9”, “S100A9” and “S100A8/A9” would be easier to understand if they were described as “S100A9/A9 homodimer”, “S100A9 monomer”, and “S100A8/A9 heterodimer” respectively.

Line 176: “2.1.1” replace by: 2.1.2

Line 206: please specify what are meant for “phosphorylated S100A8/A9” and “S100A8/PhosphoA9”. They are really not clear to me.

Line 271: please specify the acronym “TRIF”

Line 396: please generalise the numeration method for "53Cr +6.55L" and "62.9L +3,465".

Lines 649-652: some text is enlarged or italicized, apparently without reason.

Line 689: “2.4.3” replace by: 2.4.4

Author Response

In the review by Parul Singh and Syed Azmal Ali entitled “Multifunctional role of S100 protein family in the immune system: an update” the authors covered the multifaced implications of S100 protein family members in the immune system, particularly the role of S100 proteins as alarmins, antimicrobial peptides and metal-ion scavengers during innate immune responses. Further, the manuscript describes the implication of multiple S100 family proteins in the pathophysiology of COVID-19 and roles of S100A8 and S100A9 as potential prognostic markers and promising therapeutic targets. The authors also have managed to summarize a number of important features of known roles of S100 proteins in autoimmune and immune system-associated diseases. Overall, this is a useful review that aims to update what is known about the role of the S100 protein family in immune system. However, the review is somewhat confusing due to the large number of acronyms and abbreviations. A complete list of them is needed. In addition, a list of all alternative names, such as calgranulin, calprotectin, and calcyclin, would help to understand which S100 family proteins function in the context of the immune system. The following are other concerns, comments, and questions.

We would like to thank the reviewer for critical remarks and admiration of the submitted work. We have made major modifications in response to reviewer comments in order to enhance the overall quality of the paper as a result of the recommendations. We have added the complete list of acronyms and abbreviations in the revised version of the manuscript. The following are point-by-point remarks.

Line 35 and following: a doubly charged cation should be indicated as 2+ instead of +2.

Response: Thank you for pointing out for us, we have replaced all cations with 2+ by substituting +2.

Figure 1. It would be helpful to explain the domain organisation of the receptor and illustrate the binding sites of the S100 proteins to help understanding. In figure legend (line 55) and text (line 61), “G-Coupled Receptors” replace by: G protein-coupled receptors.

Response: We apologize for this typo. We have now modified and corrected the term with G protein-coupled receptors in figure as well as in main text.

Line 120 and following: it is confusing whether IL-10 affects the expression of S100A8 (lines 120-121) or not (lines 128-129) in macrophages.

Response: Apology for the mis-understanding. Yes, IL-10 induces the expression of S100A8 during bacterial LPS exposure. Initial LPS interaction with TLR-4 activates a downstream cascade. TLR-4 signalling regulates the expression of IL-10 and other class 2 transcription regulators; these transcription factors and IL10-mediated activated STAT3 induce the expression of the S100A8 monomer gene in macrophages. This leads to misunderstanding since IL-10 has a direct impact on S100A8 monomer but does not have any effect on S100A8/A9 heterodimer. The aforementioned remark has been clarified in the revised version of the text.

Line 159 and following: “S100A9/A9”, “S100A9” and “S100A8/A9” would be easier to understand if they were described as “S100A9/A9 homodimer”, “S100A9 monomer”, and “S100A8/A9 heterodimer” respectively.

Response: Thank you for your suggestion. The changes are now performed for more clarity. The complete manuscript is now corrected for respective terms “S100A9/A9 homodimer”, “S100A9 monomer”, and “S100A8/A9 heterodimer”.

Line 176: “2.1.1” replace by: 2.1.2

Response: We have corrected the heading numbering.

Line 206: please specify what are meant for “phosphorylated S100A8/A9” and “S100A8/PhosphoA9”. They are really not clear to me.

Response: S100A8/PhosphoA9 refers to the phosphorylated form of S100A9 within the S100A8/A9 complex, whereas S100A8/A9 phosphorylation is phosphorylation of both and present in the hetrodimer complex of S100A8/A9.

Line 271: please specify the acronym “TRIF”

Response: Thank you for bringing this to our attention; the acronym TIR-domain-containing adapter-inducing interferon-β (TRIF) has been added to the corresponding line.

Line 396: please generalise the numeration method for "53Cr +6.55L" and "62.9L +3,465".

Response: Thank you for your suggestion we have generalise the number for international numerical system.

Lines 649-652: some text is enlarged or italicized, apparently without reason.

Response: Thank you for highlighting. In the current version, the formatting has been rectified.

Line 689: “2.4.3” replace by: 2.4.4

Response: Yes, the numerical order was inconsistent, so we had to renumber it to put it in the proper continuous order. Now 2.4.3 is replaced with 2.3.4.

Reviewer 2 Report

In this review, the vast majority of recent studies that focused on the multifunctionality of S100 protein in the complex immune system and its associated activities has been compiled. What’s more, 25 numerous molecular approaches and signaling cascades regulated by S100 proteins during immune 26 responses and in abnormal defense systems at the time of COVID-19 pathogenesis were emphasized. Considering the hot topic of COVID-19 and the comprehensiveness of this review summary, it is recommended to be accepted by cells after minor revisions. Here are some specific suggestions:

1.      The article mentions that S100 plays an important role in the occurrence and development of various diseases. Are there any relevant studies finding that reducing S100 can reduce or delay the course of related diseases? If so, you can add relevant information to the manuscript to make the article more rigorous.

2.      S100 proteins have been demonstrated to be required by many species for their defense systems. However, the overexpression of S100 protein is accompanied by the occurrence of related diseases. Is there any research report on the balance between the two?

3.      In abstract, on line 24-25, the interaction between nerve cells and immune cells, explaining the role of S100 protein in the immune process, is not the main focus of this article, and it is not recommended to describe it in detail in the abstract section.

4.      In abstract, on line 25-27, it is suggested that in the main text, the 25 kinds of molecular pathways and signaling cascades regulated by S100 protein during the onset of COVID-19 are listed in a table.

5.      In keywords, it is recommended to select representative keywords that match the focus of the title and main text. 

6.      In introduction, S100 protein family was introduced from the aspects of structure, the involvement of cell surface receptors in various activities and functional differences, etc. It is recommended to add a paragraph to explain the review logical ideas, which would be introduced in turn from innate immune cells, antimicrobial peptides and nutritional immunity, immune system and process.

7.      Errors in some subheadings, line 176, neutrophil should be “2.1.2”, line 689, should be “2.4.4”. 

8.      In 2.3 part, functional implication of S100 protein as antimicrobial peptides and in nutritional immunity, the introduction of this part is not clear enough. It is recommended to first introduce the type of S100 protein family with antibacterial activity combined with examples. Then the antibacterial mechanism should be introduced, such as a metal scavenger in nutritional immunity, calprotectin during the formation of NETs, Zn (II) consume etc. Finally, antibacterial applications should be introduced: corneal abrasions, treatment of helicobacter pylori. 

9.      In conclusion, the three main conclusions revolve around the involvement of S100 protein in immune processes, regulation of inflammatory factors and regulation of various types of cell growth processes, it will be more prominent if it provides strategic significance for the prevention and treatment of COVID-19. 

10.  There are many writing errors in the manuscript. And the writing should be uniform throughout the manuscript, for example, Cu2+ and TNF-α are not written uniformly. The manuscript should be carefully revised.

Author Response

In this review, the vast majority of recent studies that focused on the multifunctionality of S100 protein in the complex immune system and its associated activities has been compiled. What’s more, 25 numerous molecular approaches and signaling cascades regulated by S100 proteins during immune 26 responses and in abnormal defense systems at the time of COVID-19 pathogenesis were emphasized. Considering the hot topic of COVID-19 and the comprehensiveness of this review summary, it is recommended to be accepted by cells after minor revisions. Here are some specific suggestions:

We truly appreciate the reviewer's insightful suggestions. We have taken into account every feedback and amended the paper appropriately. Revisions are underlined throughout the document in track change mode to identify where changes have made. We believe that the quality of the text has been greatly enhanced by the adjustments and enhancements suggested by the reviewers.

  1. The article mentions that S100 plays an important role in the occurrence and development of various diseases. Are there any relevant studies finding that reducing S100 can reduce or delay the course of related diseases? If so, you can add relevant information to the manuscript to make the article more rigorous.

Response: Thank you for your insightful recommendation. Yes, a member of the S100 protein family with dysregulated expression has been linked to various diseases, including cancer. Since we are concentrating on immunological prospect and its function in the immunological process. We have included information in this article about the role of S100 in relation to immune system and its dysregulation in response to immunological processes and in immune diseases. It was reported earlier that decreased level of S100 is associated with the occurrence of disease. Chronic rhinosinusitis (CRS) patients had decreased levels of S100A7 and S100A8/A9, according to research by Kim et al (Kim et al., 2019). In the context of asthma, the proinflammatory cytokines IFN-ϒ and IL-22 interact via S100A7. S100A7 is expressed in response to IL-22, which in turn inhibits IFN-ϒ (Pennino et al., 2013). Moreover, Lin et al. investigated the anti-inflammatory role of S100A8 in emphysema and found that low intracellular S100A8 levels were correlated with disease severity (Lin et al., 2019). However, expression of S100 protein members is also essential for cells surivival. S100A8 has been reported to be cytoprotective; however, knockout of this protein causes oxidative stress-induced cell apoptosis, whereas elevated levels prevent cellular injury. In OA, expression of S100A8/A9 heterodimer has been reported in the synovium and cartilage of joints. Elevation of S100A8/A9 during osteophyte formation in humans has been demonstrated by illustrating that elevated S100A8/A9 plasma levels in people with early symptomatic OA. Using S100A9 KO mice as a model for OA, author discovered that S100A8 and S100A9 are required for the formation of large osteophytes at both the bone margins and in ligaments. Previously, it was shown that cartilage damage is reduced in S100A9 KO mice during OA (Lent et al., 2012)). However, a recent study found that S100A8 and S100A9, which are important products of activated macrophages during synovial activation in OA, may increase osteophyte size in experimental OA with synovial inflammation. S100A8/A9 has the ability to upregulate and activate MMPs, which aid in cartilage matrix remodelling and allow osteophytes to grow in size (Schelbergen et al., 2016). S100A8/A9 may thus be a useful biomarker for predicting cartilage damage and osteophyte progression in human OA.

Refences:

  1. K. Kim, Y. C. Wi, S. J. Shin, K. R. Kim, D. W. Kim, and S. H. Cho, “Diverse phenotypes and endotypes of fungus balls caused by mixed bacterial colonization in chronic rhinosinusitis,” International forum of allergy & rhinology, vol. 9, no. 11, pp. 1360–1366, 2019.
  2. Pennino, P. K. Bhavsar, R. Effner et al., “IL-22 suppresses IFN-γ-mediated lung inflammation in asthmatic patients,” Journal of Allergy and Clinical Immunology, vol. 131, no. 2, pp. 562–570, 2013.
  3. R. Lin, K. Bahmed, G. J. Criner et al., “S100A8 protects human primary alveolar type II cells against injury and emphysema,” American Journal of Respiratory Cell and Molecular Biology, vol. 60, no. 3, pp. 299–307, 2019.

van Lent PL, Blom AB, Schelbergen RF, Slöetjes A, Lafeber FP, Lems WF, Cats H, Vogl T, Roth J, van den Berg WB. Active involvement of alarmins S100A8 and S100A9 in the regulation of synovial activation and joint destruction during mouse and human osteoarthritis. Arthritis Rheum. 2012 May;64(5):1466-76. doi: 10.1002/art.34315. PMID: 22143922.

Schelbergen RF, De Munter W, Van Den Bosch MH, Lafeber FP, Sloetjes A, Vogl T, et al. Alarmins S100A8/S100A9 aggravate osteophyte formation in experimental osteoarthritis and predict osteophyte progression in early human symptomatic osteoarthritis. Ann Rheum Dis (2016) 75:218–25. doi:10.1136/annrheumdis-2014-205480

  1. S100 proteins have been demonstrated to be required by many species for their defense systems. However, the overexpression of S100 protein is accompanied by the occurrence of related diseases. Is there any research report on the balance between the two?

Response: Thank you for your insight. The best example of balance between S100 protein and immunological disease condition has been discussed in 2.3.4 section. In case of pregnancy, we discussed the delicate balance between immunological processes and S100 protein, which is critical to maintain in order to avoid the adversity in pragnancy complication. In 2.3.4 section we have discusse how any alteration due to pathological conditions in cytokine release and count of myeloid cells due to any circumstances could lead to a disturbance in Th1/Th2 or pro/anti-inflammatory ratios, resulting in an alteration in the expression of S100 protein by immune non-immune cells. It resulted in altered S100 protein expression, which caused pregnancy-related complications such as embryo implantation failure, immune tolerance dysregulation, and improper decidualization or decidua formation.

For more instance, it has been reported that S100A7 expression is elevated in cases of S. aureus infection in the respiratory tract, highlighting this molecule's function in the epithelial barrier of the lower respiratory tract (Andresen et al., 2011). Moreover, recent research by Boruk et al., has confirmed that S100A9, MMP3, MMP7, MMP11, MMP25, MMP28, and CTSK protein levels are elevated in CRS nasal tissues. Proliferation of nasal epithelial cells was induced by S100A9. These findings suggest that MMP3 is sensitive to S100A9 signaling and that both molecules contribute to nasal epithelial cell proliferation (Boruk et al., 2020). More research is necessary to confirm whether S100A9 directly contributes to CRS progression. 

  1. Andresen, C. Lange, D. Strodthoff et al., “S100A7/psoriasin expression in the human lung: unchanged in patients with COPD, but upregulated upon positive S. aureusdetection,” BMC Pulmonary Medicine, vol. 11, no. 1, pp. 10–10, 2011.

Boruk, M., Railwah, C., Lora, A. et al. Elevated S100A9 expression in chronic rhinosinusitis coincides with elevated MMP production and proliferation in vitro. Sci Rep 10, 16350 (2020). https://doi.org/10.1038/s41598-020-73480-8

  1. In abstract, on line 24-25, the interaction between nerve cells and immune cells, explaining the role of S100 protein in the immune process, is not the main focus of this article, and it is not recommended to describe it in detail in the abstract section.

Response: We agree with your suggestion, and that sentence has been removed from the abstract.

  1. In abstract, on line 25-27, it is suggested that in the main text, the 25 kinds of molecular pathways and signaling cascades regulated by S100 protein during the onset of COVID-19 are listed in a table.

Response: Apologies for the inaccuracy in the sentence. In the revised version, we have modified the sentence to make it more meaningful. We have change sentence to “In addition, we have discussed the involvement of S100 protein members in abnormal defense systems during the pathogenesis of COVID-19”.

  1. In keywords, it is recommended to select representative keywords that match the focus of the title and main text. 

Response: Thank you for your suggestion; as you suggested, we have omitted the other keywords that does not appear to be relevant to our manuscript. And contain the most dominant keyword such as Nutritional immunity, inflammation, immune cells, alarmins, antimicrobial peptide, autoimmune disease, and COVID-19.

  1. In introduction, S100 protein family was introduced from the aspects of structure, the involvement of cell surface receptors in various activities and functional differences, etc. It is recommended to add a paragraph to explain the review logical ideas, which would be introduced in turn from innate immune cells, antimicrobial peptides and nutritional immunity, immune system and process.

Response: Thank you for your insightful suggestion. We have added a statements to the introduction to emphasize the importance of review. “This article highlights the multi-functional role of S100 protein members associated with the immune system. The immune system is mostly controlled by three processes and/or components that fall into two categories: innate immunity and adaptive immunity. These components include immune system cells (myeloid and lymphocytes), active molecules (alarmins, antibodies, cytokines, interleukins, chemo-attractants, antimicrobial peptides, and components of the complement system), and the immune process (inflammation, complement system, phagocytosis, and necrosis). S100 protein family members have the potential to function as active immune system molecules. In this review, we will discuss how members of the S100 protein family participate in a variety of active immunological and associated responses”.

  1. Errors in some subheadings, line 176, neutrophil should be “2.1.2”, line 689, should be “2.4.4”. 

Response: We have corrected the heading numbering.

  1. In 2.3 part, functional implication of S100 protein as antimicrobial peptides and in nutritional immunity, the introduction of this part is not clear enough. It is recommended to first introduce the type of S100 protein family with antibacterial activity combined with examples. Then the antibacterial mechanism should be introduced, such as a metal scavenger in nutritional immunity, calprotectin during the formation of NETs, Zn (II) consume etc. Finally, antibacterial applications should be introduced: corneal abrasions, treatment of helicobacter pylori

Response: Thank you for your suggestion; we have shuffled the content in response to the reviewer's suggestions.

  1. In conclusion, the three main conclusions revolve around the involvement of S100 protein in immune processes, regulation of inflammatory factors and regulation of various types of cell growth processes, it will be more prominent if it provides strategic significance for the prevention and treatment of COVID-19. 

Response: Thank you for your suggestion; we have added the content focusing on COVID-19 as you suggested. In the revised version correlation between S100A6, S100B, S100A8, S100A9, S100A12, and S100P and COVID-19 pathogenesis is discussed. In addition, an increase surge in S100A8 and S100B is associated with mild to severe COVID-19 pathogenesis. Increased levels of both proteins could be used as a biomarker for the prognosis of COVID-19 patients.

  1. There are many writing errors in the manuscript. And the writing should be uniform throughout the manuscript, for example, Cu2+ and TNF-α are not written uniformly. The manuscript should be carefully revised.

Response: Thank you for pointing out the typos; we have thoroughly reviewed the manuscript and corrected the abbreviations in their proper places.

Reviewer 3 Report

The review is devoted to proteins of the S100 family with a comprehensive coverage of processes involving it. Several technical and substantive corrections need to be made to the text:

1) Literature cited should drawn up using square brackets, inside which the numbers of sources are indicated separated by commas or dashes. For example: [3, 4]; [7-10], etc.

2) At the end of the caption to figures 2, 3, 4, 5, 6, 7, 8 the phrase “Created with BioRender.com.” write without quotes.

3) 312: instead of (S100A7, also known as psoriasis)  write (S100A7, also known as psoriasin).

4) When you are writing about AMPs (lines 303-307)  cite recent recent publication:

Guryanova, S.V.; Ovchinnikova, T.V. Immunomodulatory and Allergenic Properties of Antimicrobial Peptides. Int. J. Mol. sci. 2022, 23, 2499. https://doi.org/10.3390/ijms23052499

5) When you are writing about the influence of S100 proteins on aging, cite and discuss recent publication:

Mancinelli R, Checcaglini F, Coscia F, Gigliotti P, Fulle S, Fanò-Illic G. Biological Aspects of Selected Myokines in Skeletal Muscle: Focus on Aging. Int J Mol Sci. 2021 Aug 7;22(16):8520. doi: 10.3390/ijms22168520.

6) Source 153 is not mentioned in the text and should be removed from the bibliography.

7) In figure 6 indicate in the title of the figure that   S100 protein family member in a variety of immune-related disorders and autoimmune illnesses are abundantly expressed

8) In Figure 8, in two cases one of the symbols Th1 should be replaced by Th2.

9) In Conclusions it is written:

 Finally, the S100 proteins are required for the infection to grow and spread.

The sentence should be corrected, S100 proteins posses antimicrobial activity

Author Response

The review is devoted to proteins of the S100 family with a comprehensive coverage of processes involving it. Several technical and substantive corrections need to be made to the text:

We really appreciate the reviewer's insightful comments. All suggestions have been included into the amended article. In track change mode, changes are highlighted across the document to indicate their position. We believe that the adjustments and improvements proposed by the reviewers have significantly enhanced the text's quality.

1) Literature cited should drawn up using square brackets, inside which the numbers of sources are indicated separated by commas or dashes. For example: [3, 4]; [7-10], etc.

Response: We appreciate your suggestion and observation of the error. We have changed every multi-reference in the format that reviewer suggested.

2) At the end of the caption to figures 2, 3, 4, 5, 6, 7, 8 the phrase “Created with BioRender.com.” write without quotes.

Response: We have removed all apostrophes from the figures 2, 3, 4, 5, 6, 7, and 8.

3) 312: instead of (S100A7, also known as psoriasis)  write (S100A7, also known as psoriasin).

Response: Thank you for pointing out this spelling error; we have changed “psoriasis” to “psoriasin”.

4) When you are writing about AMPs (lines 303-307)  cite recent recent publication:

Guryanova, S.V.; Ovchinnikova, T.V. Immunomodulatory and Allergenic Properties of Antimicrobial Peptides. Int. J. Mol. sci. 2022, 23, 2499. https://doi.org/10.3390/ijms23052499

Response: Thank you for the suggestion, we have added this reference to section 2.3.

5) When you are writing about the influence of S100 proteins on aging, cite and discuss recent publication:

Mancinelli R, Checcaglini F, Coscia F, Gigliotti P, Fulle S, Fanò-Illic G. Biological Aspects of Selected Myokines in Skeletal Muscle: Focus on Aging. Int J Mol Sci. 2021 Aug 7;22(16):8520. doi: 10.3390/ijms22168520.

Response: We apologize in advance for not being able to include this reference due to a lack of related specific content in the manuscript.

6) Source 153 is not mentioned in the text and should be removed from the bibliography.

Response: Thank you for your comments; we have eliminated the reference from both the text and references.

7) In figure 6 indicate in the title of the figure that S100 protein family member in a variety of immune-related disorders and autoimmune illnesses are abundantly expressed

Response: We modified the sentence to "S100 protein family members are abundant in a variety of immune-related disorders and autoimmune illnesses." expressed

8) In Figure 8, in two cases one of the symbols Th1 should be replaced by Th2.

Response: Thank you for highlighting. We have changed the figure based on your suggestions.

9) In Conclusions it is written:

 Finally, the S100 proteins are required for the infection to grow and spread.

The sentence should be corrected, S100 proteins posses antimicrobial activity

Response: Thank you for noticing out the sentence error; as you suggested, we changed the sentence in "Finally, S100 proteins possess antimicrobial activity."

Round 2

Reviewer 3 Report

The corrections have been done. The article can be published.